# Association between First Sexual Intercourse and Sexual Violence Victimization, Symptoms of Depression, and Suicidal Behaviors among Adolescents in the United States: Findings from 2017 and 2019 National Youth Risk Behavior Survey

**DOI:** 10.3390/ijerph18157922

**Published:** 2021-07-27

**Authors:** Philip Baiden, Lisa S. Panisch, Yi Jin Kim, Catherine A. LaBrenz, Yeonwoo Kim, Henry K. Onyeaka

**Affiliations:** 1School of Social Work, University of Texas at Arlington, Arlington, TX 76019, USA; catherine.labrenz@uta.edu; 2Department of Psychiatry, University of Rochester Medical Center, Rochester, NY 14642, USA; lpanisch@utexas.edu; 3Department of Social Work, University of Mississippi, University, MS 38677, USA; yjkim@olemiss.edu; 4Department of Kinesiology, University of Texas at Arlington, Arlington, TX 76019, USA; yeonwoo.kim@uta.edu; 5Department of Psychiatry, Massachusetts General Hospital, Boston, MA 02114, USA; honyeaka@mgh.harvard.edu; 6Department of Psychiatry, Harvard Medical School, Boston, MA 02114, USA

**Keywords:** early sexual intercourse, forced sexual intercourse, sexual violence, symptoms of depression, suicidal thoughts and behaviors, adolescents

## Abstract

The objective of this study was to investigate the association between first sexual intercourse and sexual violence victimization, symptoms of depression, and suicidal ideation among sexually active adolescents in the United States. Data for this study came from the U.S. 2017 and 2019 iterations of the National Youth Risk Behavior Survey. An analytic sample of 6252 adolescents aged 14–18 years old (49.5% female) who reported ever having sexual intercourse was analyzed using Poisson regression. The outcome variables investigated in this study were sexual violence victimization, symptoms of depression, suicidal ideation, a suicide plan, and suicide attempts, and the main explanatory variables were age at first sexual intercourse and forced sexual intercourse. We also analyzed differences by gender and race. Of the 6252 adolescents who reported ever having sexual intercourse, 7.1% had their first sexual intercourse before age 13, and 14.8% experienced forced sexual intercourse. About 16% of adolescents experienced sexual violence during the past year, 42.6% reported symptoms of depression, 23.9% experienced suicidal ideation, 19.3% made a suicide plan, and 11.1% attempted suicide during the past year. In the regression analysis, early sexual intercourse was significantly and positively associated with suicidal ideation (relative risk (RR) = 1.15, 95% Confidence Interval (CI) = 1.02–1.30), suicide plan (*RR* = 1.18, 95% CI = 1.00–1.38), and suicide attempts (*RR* = 1.36, 95% CI = 1.15–1.61). Controlling for the effects of covariates, history of forced sexual intercourse was positively associated with the five outcomes examined with the relative risk ranging between 1.59 and 6.01. Findings of this study suggest that history of early or forced sexual intercourse is associated with poor mental health outcomes among adolescents and underscores the importance of developing interventions that offer psychological support in reducing the adverse impact of early sexual intercourse and forced sexual intercourse on adolescent health.

## 1. Introduction

Adolescence is a stage that is often characterized by various developmental issues, experimentation, and risk-taking [1]. Notwithstanding the fact that experiences during this stage may be positive and fulfilling, they are also likely to be stressed with intense emotions that some adolescents may not have the knowledge, experience, or skills to manage [2]. This is likely to heighten the emergence of different health risk behaviors, including early sexual intercourse (ESI) [3,4]. There is no universally acceptable definition of ESI [5]. For instance, some studies defined ESI in relation to sexual intercourse before the age of 16 [6,7], while other scholars defined ESI as sexual intercourse before the age of 14 [8]. The majority of studies, however, defined ESI to mean sexual intercourse initiated before the age of 13 [9,10,11,12,13,14] as the age of 13 marks a critical milestone in an individual’s life of transition from childhood into adolescence [15,16]. Population-based prevalence of ESI among adolescents is estimated to be in the range of 5–42%, depending on the age used in operationalizing age at ESI [17,18,19,20]. Another study that examined ESI, contraceptive use, and pregnancy outcomes among adolescents from the United States (U.S.) found that by age 13, 5.3% of adolescent males and 2.2% of adolescent females had initiated sexual intercourse [21].

A burgeoning number of studies investigating ESI have found that ESI is associated with various sexual risk behaviors such as sex without regular condom use, having multiple sexual partners, sex under the influence of alcohol or drugs, and transactional sex [22,23,24,25]. ESI has also been found to be associated with unintended pregnancy [21,26], sexually transmitted infections (STIs) [16,19,27], and increased risk of contracting human immunodeficiency virus (HIV) [28,29]. Samek et al. [30] examined data on 1512 adolescent same-sex twins from the Minnesota Twin Family Study and found that ESI was significantly associated with subsequent sexual-risk taking behaviors. Furthermore, Lara and Abdo [31] conducted a systematic review and meta-analysis of published studies on ESI and health risk behaviors among adolescent females and found that ESI was significantly and positively associated with repentance about the initiation of sexual behavior, having multiple sexual partners, STIs, and cervical cancer, and inversely associated with the use of contraceptives and self-esteem.

ESI has also been found to be associated with negative health and mental health outcomes [32,33]. For instance, among adolescents from the U.S., Ihongbe et al. [7] found that adolescents who had their sexual debut before age 12 were more than twice more likely to later experience physical dating violence when compared to their counterparts who initiated sex at age 16 years or older. Lowry et al. [34] also found that ESI before age 13 years was significantly associated with sexual risk-taking, substance use, violent victimization, and suicidal behaviors among adolescents who self-identified as lesbian, gay, or bisexual. Rector et al. [35] found that adolescents who were sexually active were more likely to experience symptoms and depression and make a suicide attempt when compared to their non-active counterparts. However, analyzing data from the National Longitudinal Study of Adolescent Health (Add Health), Sabia [36] failed to find any significant association between ESI and symptoms of depression. This was after controlling for a wide set of individual- and family-level observable factors. Using data from the Add Health study, Spriggs and Halpern [37] found that ESI was associated with depressive symptomatology among adolescent females and not adolescent males. They failed to find any significant association between ESI and depressive symptomatology by age 21 for both male and female respondents.

It is important to note that ESI may be considered as forced sexual intercourse for many adolescents or pre-adolescents, as the majority of states in the U.S. require individuals to be at least 16 years old to legally consent to sexual intercourse [38]. ESI in the context of forced sexual intercourse may also play a role in subsequent experiences of sexual violence (SV), as childhood sexual abuse has been linked to future SV victimization and perpetration [39,40]. A study examining adverse childhood experiences in relation to ESI among 241 Black females provides preliminary evidence supporting this premise [41]. Other scholars have also noted that ESI has the potential to predispose adolescents to future SV victimization [42]. Among older adolescents (i.e., those in high school) SV victimization has been linked to elevated risk of experiencing depression or trauma-related symptoms, regardless of gender or sexual orientation.

Studies that have investigated ESI have found that adolescents who engaged in ESI were more likely to experience feelings of shame and guilt [43,44,45,46]. Sprecher [46] found an inverse association between guilt and pleasure from ESI, suggesting that the greater the guilt, the less the pleasure. Other scholars have also found that adolescents who engaged in ESI reported feeling anxious about their parents finding out or were worried about getting pregnant [47]. This feeling of shame and guilt resulting from ESI could lead to symptoms of depression or engaging in suicidal behaviors as a way of numbing the shame and guilt resulting from ESI. Additionally, distressing emotions such as sex guilt may be exacerbated by sexual health interventions geared towards adolescents, which often employ shame and fear-based language in an effort to deter adolescents from early sexual activity [48], and these negative consequences may be especially amplified among adolescents with a history of forced sexual intercourse [49].

Although studies have investigated the association between ESI and risky sexual behaviors [22,23,24,25], few studies have utilized nationally representative data to examine the association between ESI, forced sexual intercourse and SV victimization, and subsequent mental health outcomes in the U.S. [7,14,34,36,37]. For instance, Lowry et al. [14] examined the association between ESI and health risk behaviors using data from the 2011 and 2013 Youth Risk Behavior Survey (YRBS) and found that ESI and forced sexual intercourse were simultaneously and independently associated with risky sexual behaviors, violence-related behaviors, and substance use. 

Our study is aimed at adding to the existing literature by using a more recent nationally representative sample of adolescents. We also sought to expand on the prior literature by using a nationally representative, more generalizable sample. The objective of this study was to investigate the association between ESI, forced sexual intercourse and SV victimization, symptoms of depression, and suicidal behaviors among adolescents who had ever had sexual intercourse using data from the 2017 and 2019 YRBS. We hypothesized that controlling for demographic factors, (1) adolescents who engaged in ESI would be at greater risk of experiencing SV, having symptoms of depression, and engaging in suicidal behaviors, and (2) adolescents with a history of forced sexual intercourse would be at greater risk of experiencing SV, having symptoms of depression, and engaging in suicidal behaviors.

## 2. Methods

### 2.1. Data Source 

Data for this study came from the Youth Risk Behavior Survey (YRBS). The YRBS is a school-based national survey conducted by the Centers for Disease Control and Prevention (CDC) every two years to examine health-risk behaviors that contribute to the leading causes of death and disability among adolescents in the U.S. The YRBS utilized a three-stage cluster sample design to recruit 9th to 12th graders from both public and private schools in the U.S. to complete self-administered surveys. Detailed information about the YRBS, including the objectives, methodology, and sampling procedure, is available in other publications [9,50,51]. The study protocol for conducting the YRBS was approved by the CDC’s Institutional Review Board (IRB), and the data is publicly available. The YRBS data were de-identified [50]; hence, no additional IRB approval was required. We followed Strengthening the Reporting of Observational Studies in Epidemiology (STROBE) guidelines in the conduct of this study [52].

### 2.2. Participants

For the purposes of this study, data from adolescent high school students from the 2017 and 2019 national YRBS were combined in order to obtain a sufficient sample size. The school response rates for the 2017 and 2019 national YRBS were 75.0% and 75.1%, respectively; student response rates were 81% and 80.3%; and overall response rates, which are a product of the school and student response rates for each cycle, were 60% for both years [9,51]. The initial sample size for the 2017 and 2019 national YRBS, respectively, were 14,765 and 13,677. The resulting sample size for the combined 2017 and 2019 national YRBS was 28,442 high school students. The sample was then restricted to adolescents aged 14 to 18 years who had ever had sexual intercourse and had complete data on all the variables included in the analysis. This resulted in a final analytic sample size of 6252 adolescents. 

### 2.3. Variables

#### 2.3.1. Explanatory Variables

The main explanatory variables investigated in this study were ESI and history of forced sexual intercourse. ESI was measured as a binary variable based on response to the question, “How old were you when you had sexual intercourse for the first time?”, with the following response options “I have never had sexual intercourse”, “11 years old or younger”, “12 years old”, “13 years old”, “14 years old”, “15 years old”, “16 years old”, and “17 years old or older.” For the purposes of this study and following the recommendation of past studies [9,10,11,12,14], adolescents who indicated having their first sexual intercourse before age 13 years old were considered as having engaged in ESI and were recoded as 1, whereas adolescents who indicated having their first sexual intercourse at aged 13 years and older were recoded as 0. Adolescents who have never had sexual intercourse were not considered in this analysis. History of forced sexual intercourse was measured as a binary variable based on response to the question, “Have you ever been physically forced to have sexual intercourse when you did not want to?” Adolescents who answered yes were considered to have experienced forced sexual intercourse and were coded as 1, otherwise they were coded as 0.

#### 2.3.2. Outcome Variables

The outcome variables investigated in this study were SV victimization, symptoms of depression, suicidal ideation, suicide plan, and suicide attempts. SV victimization was measured as a binary variable based on response to the question, “During the past 12 months, how many times did anyone force you to do sexual things that you did not want to do?” (Count such things as kissing, touching, or being physically forced to have sexual intercourse). For the purpose of this study, adolescents who experienced SV victimization at least once during the past 12 months were recoded as 1, whereas adolescents who did not experience SV victimization during the past 12 months were coded as 0. Symptoms of depression were measured as a binary variable based on response to the question, “During the past 12 months, did you ever feel so sad or hopeless almost every day for two weeks or more in a row that you stopped doing some usual activities?” Adolescents who answered “yes” were coded as 1, whereas those who answered “no” were coded as 0. Suicidal ideation was measured as a binary variable based on response to the question, “During the past 12 months, did you ever seriously consider attempting suicide?” Adolescents who answered “yes” were coded as 1, whereas adolescents who answered “no” were coded as 0. Suicide plan was measured based on response to the question, “During the past 12 months, did you make a plan about how you would attempt suicide?” Adolescents who answered “yes” were coded as 1, whereas adolescents who answered “no” were coded as 0. Suicide attempt was measured based on response to the question, “During the past 12 months, how many times did you actually attempt suicide?” Adolescents who attempted suicide at least once during the past 12 months were coded as 1, whereas adolescents who did not attempt suicide during the past 12 months were coded as 0.

#### 2.3.3. Control Variables

The following demographic factors were included in the analysis as control variables: age, gender, sexual orientation, and race/ethnicity. Age was measured in years, whereas gender was coded as “0 = Male” and “1 = Female”. Sexual orientation was coded as a nominal variable into “0 = Heterosexual”, “1 = Lesbian/gay”, “2 = Bisexual”, and “3 = Not sure”, based on sexual identity and sex of sexual contacts. Sexual orientation defined by sexual identity includes adolescents who self-identified as gay, lesbian, or bisexual and those who were not sure about their sexual identity. Sexual orientation defined by sex of sexual contacts included adolescents who had sexual contact with only the same sex or with both sexes. Race/ethnicity was coded as a nominal variable into the following categories “0 = Non-Hispanic White”, “1 = Black/African American”, “2 = Hispanic”, and “3 = Other race/ethnicity”. 

### 2.4. Data Analyses

Data were analyzed using descriptive, bivariate, and multivariate analytic techniques. The general distribution of all the variables included in the analysis was first examined using percentages. Next, we examined the bivariate association between ESI and forced sexual intercourse and the demographic variables using the Pearson Chi-square test of association. The main analysis involves the use of Poisson regression to examine the association between ESI, forced sexual intercourse, and the relative risk (RR) of the outcome variables (SV victimization, symptoms of depression, and suicidal behaviors) while simultaneously controlling for the effects of demographic variables [53,54]. Poisson regression was chosen, given that the outcome variable was measured as a binary variable and had an uneven distribution. All the variables were entered into the regression models using the enter procedure. The RRs are reported together with their 95% Confidence Intervals (CIs). Variables were considered significant if the *p*-value was less than 0.05. Stata’s “svyset” command was used to account for the weighting and complexity of the sampling design employed by the YRBS. All analyses were performed using Stata version 14.

## 3. Results

### 3.1. Sample Characteristics 

The general distribution of the study variables is presented in Table 1. Of the 6252 adolescents who reported ever having sexual intercourse, 7.1% engaged in ESI (i.e., had their first sexual intercourse before the age of 13), and about 15% reported ever experiencing forced sexual intercourse. About one in six adolescents (16%) who had ever had sexual intercourse also experienced SV during the past year and 43% reported feeling so sad or hopeless almost every day for two weeks or more in a row that they stopped doing their usual activities. With respect to suicidal behaviors, about one in four adolescents (24%) experienced suicidal ideation, 19% made a suicide plan, and 11% attempted suicide during the past year. The sample was evenly distributed by sex (female = 50.0%), and 85% self-identified as heterosexual, 10% as bisexual, 3% as lesbian/gay, and 2% as not sure.

### 3.2. Bivariate Association between Early Sexual Intercourse and Demographic Factors

As seen in Table 2, there was a significant bivariate association between ESI and forced sexual intercourse and all the demographic variables. About four out of ten (38.3%) adolescents who engaged in ESI reported ever experiencing forced sexual intercourse compared to 12.9% of adolescents who did not engage in ESI and reported ever experiencing forced sexual intercourse (χ^2^(1) = 211.14, *p* < 0.001). About one in ten males (9.2%) compared to one in twenty females (5%) engaged in ESI (χ^2^(1) = 40.32, *p* < 0.001). A little over 15% of adolescents who self-identified as lesbian/gay compared to 12.6% of bisexual, 12% of those who were not sure about their sexual orientation, and 6.1% of heterosexual engaged in ESI (χ^2^(3) = 60.50, *p* < 0.001). About 15% of adolescents who self-identified as Black/African American compared to 8.4% who self-identified as Hispanic, and 4.6% who self-identified as non-Hispanic White engaged in ESI (χ^2^(3) = 107.84, *p* < 0.001).

### 3.3. Regression Results Examining the Association between Early Sexual Intercourse and Past-year Sexual Violence Victimization, Symptoms of Depression, and Suicidal Behaviors

Table 3 shows the results of the multivariable Poisson regression examining the association of ESI and forced sexual intercourse with SV victimization, symptoms of depression, and suicidal behaviors. Controlling for the effects of forced sexual intercourse and demographic factors, we found no significant association between ESI and SV victimization and symptoms of depression. However, there was a significant association between ESI and suicidal behaviors. Controlling for the effects of forced sexual intercourse and demographic factors, adolescents who engaged in ESI had 1.15 times the risk of experiencing suicidal ideation (*RR* = 1.15, *p* = 0.027, 95% CI = 1.02–1.30), 1.18 times the risk of making a suicide plan (*RR* = 1.18, *p* = 0.05, 95% CI = 1.00–1.38), and 1.36 times the risk of having attempted suicide (*RR* = 1.36, *p* < 0.001, 95% CI = 1.15–1.61) compared to those who did not engage in ESI. Controlling for the effects of ESI and demographic factors, adolescents who reported ever experiencing forced sexual intercourse had 6.01 times the risk of experiencing SV (*RR* = 6.01, *p* < 0.001, 95% CI = 5.09–7.09), 1.59 times the risk of experiencing depressive symptoms (*RR* = 1.59, *p* < 0.001, 95% CI = 1.49–1.70), 2.09 times the risk of experiencing suicidal ideation (*RR* = 2.09, *p* < 0.001, 95% CI = 1.86–2.34), 2.22 times the risk of having made a suicide plan (*RR* = 2.22, *p* < 0.001, 95% CI = 1.94–2.54), and 2.74 times the risk of having attempted suicide (*RR* = 2.74, *p* < 0.001, 95% CI = 2.34–3.20) in comparison to their peers who had not experienced forced sexual intercourse.

Controlling for other factors, each additional increase in age was associated with a 9% reduction in the risk of SV victimization (*RR* = 0.91, *p* = 0.001, 95% CI = 0.86–0.96), a 6% reduction in the risk of symptoms of depression (*RR* = 0.94, *p* < 0.001, 95% CI = 0.92–0.97), a 6% reduction in the risk of suicidal ideation (*RR* = 0.94, *p* = 0.006, 95% CI = 0.90–0.98), a 6% reduction in the risk of suicide plan (*RR* = 0.94, *p* = 0.035, 95% CI = 0.88–1.00), a 19% reduction in the risk of suicide attempt (*RR* = 0.81, *p* < 0.001, 95% CI = 0.75–0.88). Compared to adolescent males, adolescent females had 2.49 times the risk of being a victim of SV (*RR* = 2.49, *p* < 0.001, 95% CI = 1.96–3.17), 1.59 times the risk of experiencing symptoms of depression (*RR* = 1.59, *p* < 0.001, 95% CI = 1.45–1.75), 1.36 times the risk of experiencing suicidal ideation (*RR* = 1.36, *p* < 0.001, 95% CI = 1.20–1.53). 1.28 times the risk of making a suicide plan (*RR* = 1.28, *p* = 0.001, 95% CI = 1.11–1.49), and 1.29 times the risk of making a suicide attempt (*RR* = 1.29, *p* = 0.039, 95% CI = 1.01–1.64).

## 4. Discussion

Associations between youth experiences of forced sexual intercourse and adverse mental health outcomes have been well established [55,56]. However, only a few examined the association between ESI and negative mental health outcomes [27,35,36,57,58,59], and the results are mixed [37,60]. Therefore, the objective of this study was to examine the association between ESI, forced sexual intercourse and SV victimization, symptoms of depression, and suicidal behaviors among adolescents who had ever had sexual intercourse. We found that among adolescents who reported ever having sexual intercourse, 7% engaged in ESI, while 14.8% reported ever experiencing forced sexual intercourse, and 15.5% experienced SV victimization during the past year. Regarding mental health outcomes, 42% of adolescents in the sample experienced symptoms of depression, 23.2% experienced suicidal ideation, 19.3% made a suicide plan, and 11.1% attempted suicide, all during the past year.

There is no consensus regarding the cut-off age in operationalizing ESI [5]. We conducted an area under the receiver operating characteristic curve analysis [61] to examine the sensitivity and specificity of the model for all possible cutoffs that might be used to operationalize ESI and found that the cutoff of age 13 was satisfactory. The current research found a significant association between ESI and suicidal behaviors, suggesting that adolescents who had sexual intercourse for the first time before age 13 may be more vulnerable to experience suicidal ideation, making a suicide plan, and attempting suicide compared with adolescents who initiated their first sexual intercourse at age 13 years or older. ESI was not associated with an elevated risk of depression or of past-year sexual violence victimization, although a lifetime history of forced sexual intercourse was significantly related to both variables.

The YRBS also did not ask youth to distinguish between their perceptions of ESI as forced or non-forced sexual intercourse. Rather, youth were asked to separately endorse both ESI as well as a lifetime history of forced sexual intercourse. Therefore, although our analyses sought to control for the influence of forced sexual intercourse, it is possible that there was an overlap between experiences of ESI and a lifetime history of forced sexual intercourse. Nonetheless, the results of our analyses indicate that, unlike ESI, youth’s endorsement of a lifetime history of forced sexual intercourse was an influential factor in predicting symptoms of depression, and it was also a stronger predictor than ESI for suicidal behaviors. This finding connects to literature that has examined ESI as child sexual abuse and found links between child sexual abuse and negative mental health during adolescence and young adulthood [55,56]. Other literature has conceptualized child sexual abuse as any unwanted sexual contact when the victim is a child, while others have considered child sexual abuse to be any sexual act with a child under age 16, 15, or 14, depending on the source and applicable policy or law [62]. As children under 16 are not legally able to consent to sexual intercourse [63], sexual experiences at these young ages may reflect abusive or exploitative relationships. It is important for sexual health education programs, health providers, and schools not to stigmatize or make youth feel ashamed of having experienced ESI. For example, sex education programs often employ abstinence-focused approaches [64] and use shame and fear-based language in an effort to deter adolescents from ESI [48,49]. However, this could exacerbate distressing emotions among adolescents who are already exposed to ESI, especially amongst those who have experienced SV victimization through force and/or coercion [65]. In response to this, some experts have recommended that health care professionals, especially sexuality educators, should consider developing trauma-informed comprehensive sex education programs that include protective measures for adolescents who are sexually active [66].

It may also be possible that youth’s classification of sexual experiences as either forced or not played an influential role in our findings in relation to a lifetime history of forced sexual intercourse, rather than ESI alone. The laws of consent in the U.S. would classify almost all acts of sexual intercourse before age 13 as forms of sexual violence. However, youth’s self-perceptions of agency in their early sexual experiences may have played a role in our findings. Future researchers should consider the role of sexual agency in youths’ experiences of sexual intercourse and subsequent mental health outcomes. Additionally, we encourage the use of qualitative and/or mixed-methods approaches methods to conduct a sensitive, in-depth analysis.

The current study’s findings also suggest that past-year SV victimization, symptoms of depression, and suicidal behaviors among adolescents could vary depending on demographic factors. Our findings revealed that age, being gay, lesbian, or questioning and identifying as a female were all associated with depressive symptoms and all suicidal thoughts and behaviors. Furthermore, among adolescents with a history of sexual activity, Black/African American youth had a decreased risk for depressive symptoms, suicidal ideation, and suicide plan in comparison to their White peers, while Hispanic youth had a decreased risk for suicidal ideation in comparison to White peers as well. Future research could consider these demographic factors as potential moderators in the association between ESI and suicidal thoughts and behaviors, as well as with a lifetime history of forced sexual intercourse, depressive symptoms, and suicidal thoughts and behaviors, as a childhood history of sexual violence and victimization, race/ethnicity, and both sexual orientation and gender identity have been associated with depressive symptoms and suicidal thoughts and behaviors among youth [2,67,68].

### Limitations

Despite these findings, our results should be interpreted with the following limitations in mind. First of all, although we combined the 2017 and 2019 YRBS data to obtain a sufficient sample size, there could be a possibility that same participants could answer in both the 2017 and 2019 surveys. However, because there is no personally identifiable information included in the data set, there is no way to verify how many students joined in both surveys. As previously noted, there may have been an overlap between ESI and a lifetime history of forced sexual intercourse, as well as between forced sexual intercourse and past-year SV victimization. Additionally, it should be noted that not all first sexual intercourse before age 13 years is considered to be forced sexual intercourse. Unfortunately, the YRBS does not provide detailed information to enable us to decipher whether the first sexual intercourse before age 13 years was forced sexual intercourse. This is complicated by the fact that opinions differ among researchers about what can be considered forced sexual intercourse for youth, and while lawmakers in different jurisdictions have taken a firm stance that has been incorporated into regional policies, there is an overall lack of empirical scientific evidence on the topic. Thus, these results should be interpreted with some amount of caution, as there may have been considerable overlap between a lifetime history of forced intercourse and past-year SV victimization because the YRBS did not ask youth to specify when the forced sexual intercourse occurred. In addition, the use of a cross-sectional design limits any causal interpretations of the results. It is possible that some adolescents may have experienced forced sexual intercourse or other forms of SV, depressive symptoms, or suicidal ideation prior to engaging in ESI. A longitudinal study that follows adolescents over time would help determine the temporal order between ESI and the outcome variables examined in this study. Second, this survey was conducted using self-reported measures of ESI, past-year SV victimization, and mental health, which are relatively sensitive and may be under-reported or subject to recall bias. Third, the use of secondary data limits our ability to investigate other theoretically known confounders of SV victimization and mental health outcomes such as parental history of mental illness, childhood physical abuse, and household income. In addition, although YBRS data was collected anonymously, there is still a possibility that social desirability could have biased the results, as personality and social expectations can influence even anonymous survey results. Future studies of this kind can include specific measures assessing social desirability, and these scores can then be controlled in the analysis. Lastly, we were unable to explore reasons for ESI among adolescents in the study. Future studies should utilize longitudinal designs, explore reasons for ESI, and consider these theoretically important factors to understand the true association between ESI and adverse mental health.

## 5. Conclusions

This study extends the extant literature on ESI and suggests that although ESI may lead to negative mental health outcomes in later years, a lifetime history of forced sexual intercourse is associated with increased vulnerability for negative mental health outcomes among adolescents. The findings of the current study suggest that both ESI and a lifetime history of forced sexual intercourse may predispose adolescents to suicidal thoughts and behaviors, and that forced sexual intercourse may also contribute to subsequent SV victimization and symptoms of depression. Public health initiatives that seek to address SV among adolescents should take into account a history of ESI, given its possible overlap with forced sexual intercourse and relationship with suicidal thoughts and behaviors. In accordance with recommendations of the World Health Organization, these initiatives should incorporate an ecological approach that accounts for social determinants of health at individual, relational, organizational, and community levels. Future studies should also explore the role of youth sexual self-agency in relation to possible risk factors and protective attributes. Public health interventions and trauma-informed sexual health education campaigns that seek to promote positive health behaviors might be promising avenues for enhanced developmental well-being among adolescents. 

## Figures and Tables

**Table 1 ijerph-18-07922-t001:** Sample characteristics among adolescents aged 14–18 who had ever had sex (*n* = 6252).

Variables	Frequency (Weighted %)
**Outcome variables**	
Victim of sexual violence	
No	5255 (84.0)
Yes	997 (16.0)
Experienced symptoms of depression	
No	3587 (57.4)
Yes	2665 (42.6)
Experienced suicidal ideation	
No	4757 (76.1)
Yes	1495 (23.9)
Made a suicide plan	
No	5044 (80.7)
Yes	1208 (19.3)
Attempted suicide	
No	5557 (88.9
Yes	695 (11.1)
**Explanatory variables**	
Engaged in early sexual intercourse	
No	5806 (92.9)
Yes	446 (7.1)
Ever experienced forced sexual intercourse	
No	5330 (85.2)
Yes	922 (14.8)
**Demographic and control variables**	
Age	
14 years	264 (4.2)
15 years	965 (15.4)
16 years	1631 (26.1)
17 years	2101 (33.6)
18 years	1291 (20.7)
Sex	
Male	3157 (50.5)
Female	3095 (49.5)
Sexual orientation	
Heterosexual	5323 (85.2)
Lesbian/gay	163 (2.6)
Bisexual	623 (10.0)
Unsure	143 (2.2)
Race/ethnicity	
Non-Hispanic White	3428 (54.8)
Black/African American	667 (10.7)
Hispanic	1615 (25.8)
Other race/ethnicity	542 (8.7)

**Table 2 ijerph-18-07922-t002:** Bivariate association between demographic factors and early sexual intercourse among adolescents aged 14–18 who had ever had sex (*n* = 6252).

Variables	Engaged in Early Sexual Intercourse (Weighted %)		Chi-Square (df), *p*-Value
	No	Yes	
Ever experienced forced sexual intercourse			211.14 (1), *p* < 0.001
No	87.1	61.7	
Yes	12.9	38.3	
Age			173.75 (4), *p* < 0.001
14 years	77.6	22.4	
15 years	87.8	12.2	
16 years	92.3	7.7	
17 years	95.7	4.3	
18 years	95.9	4.1	
Sex			40.32 (1), *p* < 0.001
Male	90.8	9.2	
Female	95.0	5.0	
Sexual orientation			60.50 (3), *p* < 0.001
Heterosexual	93.9	6.1	
Lesbian/gay	84.1	15.9	
Bisexual	87.4	12.6	
Unsure	88.0	12.0	
Race/ethnicity			107.84 (3), *p* < 0.001
Non-Hispanic White	95.4	4.6	
Black/African American	84.9	15.1	
Hispanic	91.6	8.4	
Other	90.1	9.9	

Note: df, degrees of freedom.

**Table 3 ijerph-18-07922-t003:** Association between ESI and *past 12 months* SV victimization, symptoms of depression, and suicidal ideation among adolescents aged 14–18 who had ever had sex (*n* = 6252).

Variables	Sexual Violence Victimization		Symptoms of Depression		Suicidal Ideation		Suicide Plan		Suicide Attempts	
	RR (95% CI)	*p*-Value	RR (95% CI)	*p*-Value	RR (95% CI)	*p*-Value	RR (95% CI)	*p*-Value	RR (95% CI)	*p*-Value
Engaged in early sexual intercourse (No)										
Yes	1.00 (0.82–1.22)	0.989	1.05 (0.95–1.15)	0.343	1.15 (1.02–1.30)	0.027	1.18 (1.00–1.38)	0.050	1.36 (1.15–1.61)	<0.001
Ever experienced forced sexual intercourse (No)										
Yes	6.01 (5.09–7.09)	<0.001	1.59 (1.49–1.70)	<0.001	2.09 (1.86–2.34)	<0.001	2.22 (1.94–2.54)	<0.001	2.74 (2.34–3.20)	<0.001
**Demographic variables**										
Age in years	0.91 (0.86–0.96)	0.001	0.94 (0.92–0.97)	<0.001	0.94 (0.90–0.98)	0.006	0.94 (0.88–1.00)	0.035	0.81 (0.75–0.88)	<0.001
Sex (Male)										
Female	2.49 (1.96–3.17)	<0.001	1.59 (1.45–1.75)	<0.001	1.36 (1.20–1.53)	<0.001	1.28 (1.11–1.49)	0.001	1.29 (1.01–1.64)	0.039
Sexual orientation (Heterosexual)										
Lesbian/gay	0.58 (0.38–0.87)	0.010	1.46 (1.25–1.71)	<0.001	2.30 (1.83–2.89)	<0.001	2.22 (1.72–2.87)	<0.001	1.95 (1.37–2.78)	<0.001
Bisexual	1.08 (0.92–1.27)	0.343	1.41 (1.30–1.52)	<0.001	2.19 (1.94–2.47)	<0.001	2.24 (1.93–2.60)	<0.001	2.30 (1.81–2.91)	<0.001
Unsure	1.55 (1.24–1.94)	<0.001	1.59 (1.39–1.83)	<0.001	2.30 (1.89–2.81)	<0.001	2.43 (1.87–3.15)	<0.001	2.35 (1.73–3.19)	<0.001
Race/ethnicity (non-Hispanic White										
Black/African American	0.70 (0.55–0.90)	0.005	0.80 (0.72–0.90)	<0.001	0.72 (0.59–0.88)	0.002	0.74 (0.58–0.94)	0.015	0.96 (0.73–1.26)	0.760
Hispanic	0.99 (0.86–1.10)	0.630	1.06 (0.98–1.15)	0.167	0.85 (0.76–0.96)	0.010	0.98 (0.82–1.16)	0.803	1.07 (0.85–1.36)	0.553
Other	1.04 (0.85–1.29)	0.685	1.10 (0.99–1.23)	0.089	1.07 (0.89–1.27)	0.475	1.32 (1.10–1.59)	0.004	1.32 (1.10–1.58)	0.003

Note: Reference category is indicated in parenthesis. RR = relative risk; CI = Confidence Interval.

## Data Availability

This research is based on publicly available data from the 2017 and 2019 Youth Risk Behavior Survey (YRBS), collected by the Centers for Disease Control and Prevention (CDC). This data can be found at: https://www.cdc.gov/healthyyouth/data/yrbs/index.htm (accessed on 21 August 2020). The views and opinions expressed in this paper are those of the authors and do not necessarily represent the views of CDC or that of its partners. Dr. Baiden had full access to the data and takes responsibility for the integrity of the data and the accuracy of the data analysis. The data that support the findings of this study are available from the corresponding author, upon reasonable request.

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
