# Peer review of "Association between First Sexual Intercourse and Sexual Violence Victimization, Symptoms of Depression, and Suicidal Behaviors among Adolescents in the United States: Findings from 2017 and 2019 National Youth Risk Behavior Survey"

_ijerph, 2021, doi:10.3390/ijerph18157922_

Round 1

Reviewer 1 Report

Report on the article titled “Association between first sexual intercourse and sexual violence victimization, symptoms of depression, and suicidal behavior among adolescent: Findings from 2017 and 2019 national youth risk behavior survey”.

Brief summary

This study examined the association between first sexual intercourse/early sexual intercourse and sexual violence victimization, depression, suicidal ideation, suicidal plan, and attempt.

Strength of the study
This is a national representative study which used national data in the analyses section

Weaknesses of the study
The variables and indicators of the study were not resonated in all part of the study. Thus, making the results, discussion of findings, and conclusion very clumsy and difficult to comprehend. There should be consistency between the results and discussion of findings, which should culminate to the conclusion. For example, how was depression measured? What are the symptoms of depression as used in this study? These variables need to resonate frequently while discussing your findings in the light of early sexual intercourse in the discussion of finding section. The readers do not need to get lose throughout the study. In addition, it is not all first sexual intercourse before age 13 years that was forced. Kindly filter your dataset to make the analyses unique. If the dataset does not provide this differentiation, kindly state it as a limitation in the limitation of the study section.

Methodology section
1. What is YRBS? State in full
2. Kindly describe the study location
3. Kindly state reasons for the statistical methods utilized in the study

Results
1. There is a disjunction between the results and the discussion of finding section. The analyses (see table 2) highlights between demographics (Age 14-18 years) and early sexual intercourse. However, the conceptualization of the discussion was before 13 years or after 13 years (see line 298-305). Also see line 155-159. These discrepancies are very significant. Kindly align your conceptualization and be consistent with them.
2. The sample was not evenly distributed as claim in line 224. There is no half human being, so 49.5% is not an even distribution. Kindly approximate to 50%
3. Kindly state the implications of the results

Other comments
1. The first paragraph in the discussion section (line 279-297) should be expunch and taken to the background/introduction section
2. Line 308-311 should be taken to limitation of the study
3. The use of acronym such as CSA should be written in full (see line 321)
4. Delete the sentence “in other studies” in line 327. It makes the paragraph clumsy
5. Sentences are too long (337-343). Kindly use short, sharp, and clear sentences.
6. Kindly critique line 354-356 to understand why such outcome was outstanding in your data.
7. Line 390….why are you introducing mental health at the conclusion section? Was this study about mental health? Kindly expunch.
8. The variables sexual violence, depression, and suicidal ideation should be thoroughly interrogated in the discussion of finding section
9. This study needs thorough language editing and plagiarism check
10. Avoid clumsy and long sentences.

Author Response

Response to Comments from Reviewer #1

This study examined the association between first sexual intercourse/early sexual intercourse and sexual violence victimization, depression, suicidal ideation, suicidal plan, and attempt.

1.1 Strength of the study
This is a national representative study which used national data in the analyses section.

Response: We thank the reviewer for this observation.

1.2 Weaknesses of the study
The variables and indicators of the study were not resonated in all part of the study. Thus, making the results, discussion of findings, and conclusion very clumsy and difficult to comprehend. There should be consistency between the results and discussion of findings, which should culminate to the conclusion. For example, how was depression measured? What are the symptoms of depression as used in this study? These variables need to resonate frequently while discussing your findings in the light of early sexual intercourse in the discussion of finding section. The readers do not need to get lose throughout the study. In addition, it is not all first sexual intercourse before age 13 years that was forced. Kindly filter your dataset to make the analyses unique. If the dataset does not provide this differentiation, kindly state it as a limitation in the limitation of the study section.

Response: This is a valid point to note and we thank the reviewer for raising this. On page 10, we do recognize that, although legally children under age 13 cannot consent in most states to sexual intercourse, youth perception of agency may serve as a protective factor. Unfortunately, the YRBS does not provide detailed information to enable us to decipher the first sexual intercourse before age 13 years was forced sexual intercourse. This has been acknowledged as a limitation in the revised manuscript. The manuscript has also been carefully checked for consistency with the use of terminologies.

Methodology section
1.3. What is YRBS? State in full

Response: This has been stated on page 3 of the revised manuscript. The sentence now reads: “Data for this study came from the Youth Risk Behavior Survey (YRBS).”

1.4. Kindly describe the study location

Response: The data were gathered on adolescents in 9th to 12th grades from both public and private schools in the U.S. The sentence on page 3 of the revised manuscript now reads:

The YRBS utilized a three-stage cluster sample design to recruit 9th to 12th graders from both public and private schools in the U.S. to complete self-administered surveys.

1.5. Kindly state reasons for the statistical methods utilized in the study

Response: We thank the reviewer for this question. The following has been added to page 5 of the revised manuscript: “Poisson regression was chosen given that the outcome variable was measured as a binary variable and had an uneven distribution.”

Results
1.6. There is a disjunction between the results and the discussion of finding section. The analyses (see table 2) highlights between demographics (Age 14-18 years) and early sexual intercourse. However, the conceptualization of the discussion was before 13 years or after 13 years (see line 298-305). Also see line 155-159. These discrepancies are very significant. Kindly align your conceptualization and be consistent with them.

Response:

1.7. The sample was not evenly distributed as claim in line 224. There is no half human being, so 49.5% is not an even distribution. Kindly approximate to 50%.

Response: We have approximated the percentages describing our sample.

1.8. Kindly state the implications of the results

Response: On page 10, we have lines 346-369, we include implications for practice and for research.

Other comments
1.9. The first paragraph in the discussion section (line 279-297) should be expunch and taken to the background/introduction section
Response: We thank the reviewer for this comment. We have revised the first paragraph of the discussion based on the reviewer’s suggestion.

1.10. Line 308-311 should be taken to limitation of the study

Response: The sentence in line 308-311 has been moved to the limitation.

1.11. The use of acronym such as CSA should be written in full (see line 321)

Response: The acronym CSA has been spelled out in full throughout the manuscript as child sexual abuse

1.12. Delete the sentence “in other studies” in line 327. It makes the paragraph clumsy

Response: The sentence in line 327 has been deleted.

1.13. Sentences are too long (337-343). Kindly use short, sharp, and clear sentences.

Response: The sentences in line 337-343 have been revised based on the suggestion from the reviewer.

1.14. Kindly critique line 354-356 to understand why such outcome was outstanding in your data.

Response: The sentence in lines 354-356 has been revised and now reads: “Furthermore, among adolescents with a history of sexual activity, Black/ African American youth had a decreased risk than their White peers for depressive symptoms, suicidal ideation, and making a suicide plan, while Hispanic youth also had a decreased risk for suicidal ideation than their White counterparts.”

1.15. Line 390….why are you introducing mental health at the conclusion section? Was this study about mental health? Kindly expunch.

Response: The sentence in line 390 has been revised and now reads: “Public health initiatives that seek to address SV among adolescents should take into account a history of ESI, given its possible overlap with forced sexual intercourse, and relationship with suicidal thoughts and behaviors.”

1.16. The variables sexual violence, depression, and suicidal ideation should be thoroughly interrogated in the discussion of finding section

Response: We thank the reviewer for raising these points. However, we wish to highlight that in keeping with the scope and word of the manuscript, we have now revised and provided an appropriate discussion of the findings in our discussion.

1.17. Avoid clumsy and long sentences.

Response: The manuscript has also been thoroughly checked for clumsy and long sentences.

Reviewer 2 Report

Very interesting paper. Please include the following questions in the text:

  • The country must be indicated in the title and abstract. The reader does not know which country the data come from until he/she has advanced in the reading.
  • Differences by gender and ethnicity are obtained. Note this in the abstract.
  • Report whether the 2017 and 2019 samples are independent, or there may be adolescents who responded to both.
  • Social desirability bias also occurs in anonymous responses. It is related to personality and social expectations. To assess its effect, there are specific social desirability questionnaires whose scores can be controlled in the analyses. Please delete this sentence: "The YRBS data were collected anonymously; hence there is no worry about social desirability bias" (line 132-133). Add this remark in limitations.

DISCUSSION:  

The authors differentiate between ESI and forced sexual relations. They also use the concept of "sexual agency" for children or adolescents. Other authors consider that all sexual relations with no "sexually adults" are forced, due to the lack of sexual maturity, and only differentiate between "violent" (forced for the authors) or seduced (ESI for the authors). Since there is no evidence to support these speculations, please explain clearly that this is an unscientific opinion, or suppress opinions not supported by empirical evidence, so as not to confuse readers. It is important that the inferences of a research study are limited to the conclusions that the data allow to draw, in order to assecurate the study validity (AERA, NCME, APA, 2014)

The same observation for the recommendation of "not stigmatizing" ESI in sex education programs: "As such, it is important for sexual health education programs, health providers, and schools not to stigmatize ESI" (lines 328-329). Do the authors suggest that the harmful effects of ESI is its stigmatization? What does "no stigmatization" mean? Do they mean that ESI should be normalized? Or do they mean that sex education programs should also offer complementary programs for those with ESI experiences? 

The paper provides important research findings. The results are of interest to other researchers. Limiting the conclusions strictly to the results will improve the validity of the report.

Author Response

Response to Comments from Reviewer #2

Very interesting paper. Please include the following questions in the text:

2.1. The country must be indicated in the title and abstract. The reader does not know which country the data come from until he/she has advanced in the reading.

 Response: We thank the reviewer for pointing this out and have indicated the name of the country (the United States) in the title and abstract.

2.2. Differences by gender and ethnicity are obtained. Note this in the abstract.

 Response: The sentence in line 390 has been revised and now reads: “Public health initiatives that seek to address SV among adolescents should take into account a history of ESI, given its possible overlap with forced sexual intercourse, and relationship with suicidal thoughts and behaviors.”

2.3. Report whether the 2017 and 2019 samples are independent, or there may be adolescents who responded to both.

 Response: Thank you for your valuable comment. It could be a possible limitation of the study because same participants could answer in both 2017 and 2019 survey. So, we added this issue in the limitation section: “First of all, although we combined the 2017 and 2019 YRBS data to obtain a sufficient sample size, there could be a possibility that same participants could answer in both 2017 and 2019 survey. However, since there is no personally identifiable information included in the data set, there is no way to verify how many students joined in both surveys.” (p. 10, lines 489-493)

2.4. Social desirability bias also occurs in anonymous responses. It is related to personality and social expectations. To assess its effect, there are specific social desirability questionnaires whose scores can be controlled in the analyses. Please delete this sentence: "The YRBS data were collected anonymously; hence there is no worry about social desirability bias" (line 132-133). Add this remark in limitations.

 Response: We appreciate the reviewer’s feedback. We have deleted this sentence from the Methods, and have added the suggested comments to the limitations section:

“In addition, although YBRS data was collected anonymously, there is still a possibility that social desirability could have biased the results, as personality and social expectations can influence even anonymous survey results. Future studies of this kind can include specific measures assessing social desirability, and these scores can then be controlled in the analysis.”

DISCUSSION:  

2.5. The authors differentiate between ESI and forced sexual relations. They also use the concept of "sexual agency" for children or adolescents. Other authors consider that all sexual relations with no "sexually adults" are forced, due to the lack of sexual maturity, and only differentiate between "violent" (forced for the authors) or seduced (ESI for the authors). Since there is no evidence to support these speculations, please explain clearly that this is an unscientific opinion, or suppress opinions not supported by empirical evidence, so as not to confuse readers. It is important that the inferences of a research study are limited to the conclusions that the data allow to draw, in order to assecurate the study validity (AERA, NCME, APA, 2014).

Response: We have added the following to the Limitations section:

“This is complicated by the fact that opinions differ among researchers about what can be considered forced sexual intercourse for youth and while lawmakers in different jurisdictions have taken a firm stance that has been incorporated into regional policies, there is an overall lack of empirical scientific evidence on the topic.”

2.6. The same observation for the recommendation of "not stigmatizing" ESI in sex education programs: "As such, it is important for sexual health education programs, health providers, and schools not to stigmatize ESI" (lines 328-329). Do the authors suggest that the harmful effects of ESI is its stigmatization? What does "no stigmatization" mean? Do they mean that ESI should be normalized? Or do they mean that sex education programs should also offer complementary programs for those with ESI experiences? 

Response: We appreciate these questions from the reviewer. To explain stigmatization and it’s harmful effects, we wrote in that section: “sex education programs often employ abstinence-focused approaches [64] and use shame and fear-based language in an effort to deter adolescents from ESI [48,49]. However, this could exacerbate distressing emotions among adolescents who are already exposed to ESI especially amongst those who have experienced SV victimization through force and/or coercion [65].”

We do not mean that ESI should be normalized or that sex education programs should also offer complementary programs for those with ESI experiences. Rather, as we have written, we agree with suggestions of experts in this area who recommend that “health care professionals, especially sexuality educators, should consider developing trauma-informed comprehensive sex education programs that include protective measures for adolescents who are sexually active [66].”

2.7. The paper provides important research findings. The results are of interest to other researchers. Limiting the conclusions strictly to the results will improve the validity of the report.

Response: We thank the reviewer for this positive feedback.

Reviewer 3 Report

Type of manuscript: Review

Manuscript ID: ijerph-1286148
Title: Association between First Sexual Intercourse and Sexual Violence Victimization, Symptoms of Depression, and Suicidal Behaviors among Adolescents: Findings from 2017 and 2019 National Youth Risk Behavior Survey
Submitted to section: Children's Health,

I review Manuscript ID entitled "Association between First Sexual Intercourse and Sexual Violence Victimization, Symptoms of Depression, and Suicidal Behaviors among Adolescents: Findings from 2017 and 2019 National Youth Risk Behavior Survey”

The aim of the study was to investigate the association between first sexual intercourse and sexual violence victimization, symptoms of depression, and suicidal ideation among sexually active adolescents. Data for this study came from the 2017 and 2019 iterations of the National Youth Risk Behavior Survey.

It is a very interesting topic, however, the paper , in my opinion, needs some revisions.

TITLE: SOUND

Abstract SOUND but I would try to better describe the gap in the literature that the authors intend to fill with their study.

Introduction The literature review is an important part in this article so, in the Introduction, the gap in literature should be better explained to better understand the importance of your study.

This part could be better structured. For example I would have put in the Introduction some study on how intimate relationships are structured in adolescence and emerging adults (Arnett, J.J. Emerging adulthood: A theory of development from the late teens through the twenties. Am. Psychol. 2000, 55, 469–480) more generally. I would have more "stressed" the gap that exists in the literature and how your study proposes to fill it. Then, I would have added a specific paragraph on ESI (about the part on ESI I think it is great and complete).

I would also suggest to make a reference to the studies that relate attachment to parents, love style and depression in young people (e.g. references about attachment to parents, love style, and depression).

Meadows, A. L., Coker, A. L., Bush, H. M., Clear, E. R., Sprang, G., & Brancato, C. J. (2020). Sexual violence perpetration as a risk factor for current depression or posttraumatic symptoms in adolescents. Journal of interpersonal violence.

Fermani, A., Bongelli, R., Canestrari, C., Muzi, M., Riccioni, I., & Burro, R. (2020). “Old Wine in a New Bottle”. Depression and Romantic Relationships in Italian Emerging Adulthood: The Moderating Effect of Gender. International Journal of Environmental Research and Public Health,17(11), 4121;

Bifulco, A., Jacobs, C., Bunn, A., Damiani, R., & Spence, R. (2019). Partner violence in women: associations with childhood maltreatment, attachment style and Major Depression. Partner violence in women: associations with childhood maltreatment, attachment style and Major Depression, 13-28.

Del Moral, Gonzalo; Franco, Cosette; Cenizo, Manuel; Canestrari, Carla; Suárez-Relinque, Cristian; Muzi, Morena; Fermani, Alessandra. 2020. "Myth Acceptance Regarding Male-To-Female Intimate Partner Violence amongst Spanish Adolescents and Emerging Adults". Int. J. Environ. Res. Public Health,17, 21: 8145.)

(Lines 112- 120) in my opinion the aim and hypothesis should be better explained in a separate paragraph.

Method SOUND

Results SOUND

Discussion, Conclusion, Limitations and References

Very good!!!

I would remind to the authors the importance of keeping in mind and underlining an ecological approach (an individual, relational, organizational, and community levels) as pointed out by the WHO (2012) and by recent papers.

Di Napoli, I., Procentese, F., Carnevale, S., Esposito, C., & Arcidiacono, C. (2019). Ending intimate partner violence (IPV) and locating men at stake: an ecological approach. International journal of environmental research and public health, 16(9), 1652.

Eventually add all the references indicated above, please.

The study is very interesting and I really enjoyed the methodological part and the discussions. I would, therefore, like to congratulate your entire research team.

Best regards

Author Response

Response to Comments from Reviewer #3

Review Manuscript ID entitled "Association between First Sexual Intercourse and Sexual Violence Victimization, Symptoms of Depression, and Suicidal Behaviors among Adolescents: Findings from 2017 and 2019 National Youth Risk Behavior Survey”

The aim of the study was to investigate the association between first sexual intercourse and sexual violence victimization, symptoms of depression, and suicidal ideation among sexually active adolescents. Data for this study came from the 2017 and 2019 iterations of the National Youth Risk Behavior Survey.

It is a very interesting topic, however, the paper , in my opinion, needs some revisions.

 TITLE: SOUND

Abstract 

3.1. SOUND but I would try to better describe the gap in the literature that the authors intend to fill with their study.

 Response: We added the following sentence on page 3 to describe the gap that this study fills: “Our study is aimed at adding to the existing literature by using a more recent nationally representative sample of adolescents. We also sought to expand on the prior literature by using a nationally representative, more generalizable sample.”

Introduction 

3.2. The literature review is an important part in this article so, in the Introduction, the gap in literature should be better explained to better understand the importance of your study.

 This part could be better structured. For example I would have put in the Introduction some study on how intimate relationships are structured in adolescence and emerging adults (Arnett, J.J. Emerging adulthood: A theory of development from the late teens through the twenties. Am. Psychol. 2000, 55, 469–480) more generally. I would have more "stressed" the gap that exists in the literature and how your study proposes to fill it. Then, I would have added a specific paragraph on ESI (about the part on ESI I think it is great and complete).

Response:

3.3. I would also suggest to make a reference to the studies that relate attachment to parents, love style and depression in young people (e.g. references about attachment to parents, love style, and depression).

Meadows, A. L., Coker, A. L., Bush, H. M., Clear, E. R., Sprang, G., & Brancato, C. J. (2020). Sexual violence perpetration as a risk factor for current depression or posttraumatic symptoms in adolescents. Journal of interpersonal violence.

 Fermani, A., Bongelli, R., Canestrari, C., Muzi, M., Riccioni, I., & Burro, R. (2020). “Old Wine in a New Bottle”. Depression and Romantic Relationships in Italian Emerging Adulthood: The Moderating Effect of Gender. International Journal of Environmental Research and Public Health,17(11), 4121;

 Bifulco, A., Jacobs, C., Bunn, A., Damiani, R., & Spence, R. (2019). Partner violence in women: associations with childhood maltreatment, attachment style and Major Depression. Partner violence in women: associations with childhood maltreatment, attachment style and Major Depression, 13-28.

 Del Moral, Gonzalo; Franco, Cosette; Cenizo, Manuel; Canestrari, Carla; Suárez-Relinque, Cristian; Muzi, Morena; Fermani, Alessandra. 2020. "Myth Acceptance Regarding Male-To-Female Intimate Partner Violence amongst Spanish Adolescents and Emerging Adults". Int. J. Environ. Res. Public Health,17, 21: 8145.)

 Response: We appreciate the reviewer’s suggestion. However, the relationships between attachment to parents, love style, and depression among youth are beyond the scope of this article and our outcomes under investigation. Nonetheless, we truly appreciate the suggested articles the reviewer has generously provided, and have included information from the findings of Meadows et al., which are indeed relevant to our study premise:

Meadows et al. (2020): “Among older adolescents (i.e., those in high school) SV victimization has been linked to elevated risk of experiencing depression or trauma-related symptoms, regardless of gender or sexual orientation.”

3.4. (Lines 112- 120) in my opinion the aim and hypothesis should be better explained in a separate paragraph.

 Response: We have revised the text so that the aim and hypothesis are now in a separate paragraph.

Method SOUND

 Results SOUND

 Discussion, Conclusion, Limitations and References

Very good!!!

3.5. I would remind to the authors the importance of keeping in mind and underlining an ecological approach (an individual, relational, organizational, and community levels) as pointed out by the WHO (2012) and by recent papers.

Di Napoli, I., Procentese, F., Carnevale, S., Esposito, C., & Arcidiacono, C. (2019). Ending intimate partner violence (IPV) and locating men at stake: an ecological approach. International journal of environmental research and public health, 16(9), 1652.

Response: We appreciate this suggestion and have added the following to our Conclusions:

“In accordance with recommendations of the World Health Organization, these initiatives should incorporate an ecological approach that accounts for social determinants of health at individual, relational, organizational, and community levels (Di Napoli et al., 2019).”

3.6. Eventually add all the references indicated above, please.

 Response: We thank the reviewer for generously providing is with these suggested references and we have added all references that are relevant to the scope of the study.

3.7 The study is very interesting and I really enjoyed the methodological part and the discussions. I would, therefore, like to congratulate your entire research team.

Best regards

Response: We thank the reviewer for the favorable assessment of our manuscript.